# Fractional Criticality Theory and Its Application in Seismology

**Boris Shevtsov** *,† [ID] **and Olga Sheremetyeva** † [ID]

Institute of Cosmophysical Research and Radio Wave Propagation, Far Eastern Branch
of the Russian Academy of Sciences, Mirnaya Str. 7, Elizovskiy District, 684034 Paratunka, Kamchatka Region,
Russia; sheremeteva@ikir.ru
* Correspondence: bshev@ikir.ru
† These authors contributed equally to this work.

**Abstract:** To understand how the temporal non-locality («memory») properties of a process affect its critical regimes, the power-law compound and time-fractional Poisson process is presented as a universal hereditary model of criticality. Seismicity is considered as an application of the theory of criticality. On the basis of the proposed hereditarian criticality model, the critical regimes of seismicity are investigated. It is shown that the seismic process has the property of «memory» (non-locality over time) and statistical time-dependence of events. With a decrease in the fractional exponent of the Poisson process, the relaxation slows down, which can be associated with the hardening of the medium and the accumulation of elastic energy. Delayed relaxation is accompanied by an abnormal increase in fluctuations, which is caused by the non-local correlations of random events over time. According to the found criticality indices, the seismic process is in subcritical regimes for the zero and first moments and in supercritical regimes for the second statistical moment of events' reoccurrence frequencies distribution. The supercritical regimes indicate the instability of the deformation changes that can go into a non-stationary regime of a seismic process.

**Keywords:** hereditary theory of criticality; critical phenomena; compound fractional Poisson process; scaling of random event streams; critical indices; critical regimes; coherent effects; deformation theory

**MSC:** 33E12; 60G22; 60G55

## 1. Introduction

Critical phenomena may have different natures, but they are united by three common properties: scaling of the stream of events and a power-law divergence of the process characteristics near critical points, while in the dynamics of the process there is a slow-down before explosive activation. It will be shown below that, if the compound Poisson process is provided with such properties, then it will be a completely adequate model of a critical phenomenon.

The scaling of the event stream is given by a power-law distribution, the non-integrability of which on small or large scales leads to divergences in the statistical characteristics of the process. A special case of logarithmic divergence simultaneously on small and large scales occurs when the exponent is minus one (the distribution of $(1/x)$ is not integrated at both ends). This value can be considered the most important critical index.

One consolidation of scales according to the power-law is not enough for a complete understanding of the nature of the critical phenomenon. No less important is the temporal correlation of events, for which the «memory» of the process is responsible. It will be shown below that scaling and hereditarity are implemented multiplicatively, and how the behavior of the statistical characteristics of the process depends on it.

We consider the compound Poisson process [1,2] in its fractional generalization [3–5] with a power-law distribution of events' recurrence frequencies [6,7]. Due to scaling and hereditarity, this process has all the necessary properties with which to describe

critical phenomena, including seismic ones. Scaling generates the divergence of statistical moments, and hereditarity generates the statistical dependence of random events, delayed relaxations, and growth of fluctuations. Divergences of statistical moments cause instability of deformation changes, which can unexpectedly switch to a non-stationary regime of the seismic process activation, and hereditarity increases the determinacy of changes. The slowdown in relaxation is related to the medium hardening and the energy accumulation, and the growth of fluctuations indicates collective effects caused by the consolidation of spatial and temporal scales.

We do not limit the applications of the hereditarian theory of criticality to seismology only, but use it only as one of the examples. The results obtained here can be applied to describe critical phenomena of any other nature.

In the next section, we will define the compound fractional Poisson process [3–5] and provide it with a power-law of the distribution of the amplitudes of changes, i.e., scaling [6,7], and thus obtain the power-law compound fractional Poisson process. In Section 3, we will define the critical indices and regimes of this process, which will allow us to consider it as a fairly universal model of the hereditarian theory of criticality. In Section 4, we will consider the application of the hereditarian criticality model to the study of the critical indices and regimes of the seismic process using the example of seismic data from the earthquake catalog of the Kamchatka Branch of the Geophysical Survey of the Russian Academy of Sciences [8]. In Section 5, we will collect and discuss the results and, in Section 6, we will present conclusions based on them.

## 2. Compound Fractional Poisson Process with Power-Law Distribution of Events Recurrence Frequencies

### 2.1. Compound Fractional Poisson Process

The seismic process is diverse and can have different representations; we are considering one of its possible representations. The seismic process is heterogeneous and non-stationary, which means that the event stream density depends on coordinates and time. If we average the event stream density over the area of event coordinates and the observation time, we can represent this process as quasi-homogeneous and quasi-stationary. To describe this process, we use the compound fractional Poisson process (CFPP) of order $k$ with integer random state changes by $r = 1, 2, \ldots, k$ and positive rates $\lambda_r$ as it is represented in the papers [3,4]:

$$\frac{d^\nu}{dt^\nu} p_\nu(j,t) = -\Lambda p_\nu(j,t) + \sum_{r=1}^{min\{j,k\}} \lambda_r p_\nu(j-r,t), \quad j \in \mathbb{N}_0, \quad \Lambda = \sum_{r=1}^{k} \lambda_r, \tag{1}$$

with initial conditions

$$p_\nu(j,0) = \begin{cases} 1, & j = 0, \\ 0, & j \geq 1, \end{cases}$$

where the time $t \geq 0$, the $p_\nu(j,t)$ is probability of the process to be in one of the possible states $j$, and the $\nu$ is the exponent of the Caputo fractional derivative, $0 < \nu \leq 1$.

### 2.2. Distribution of Recurrence Frequencies of Events

The distribution of positive rates $\lambda_r$ (1) in the case of a seismic process is defined by the Gutenberg–Richter law for magnitudes $1 < M < 9$ [6],

$$N(m \geq M) = 10^{a-bM} = N_{total} \, 10^{-bM}, \tag{2}$$

where $N_{total} = 10^a$ is the total number of events.

There are various definitions of magnitude. In our study, we will use the definition of Kanamori [7],

$$M = (2/3)(\lg M_0 - C), \tag{3}$$

where $M_0$ is the seismic moment and $C$ is the normalization constant depending on the seismic moment $m_0$ of the calibration event

$$C = \lg m_0, \tag{4}$$

the energy of which we will consider to be minimal. Using other definitions of magnitude will require another renormalization of the event energy, but will not change the power-law events distribution by the energy.

Given the dependencies $M_0 = \mu S u$ [7], $S = L^2$ and $u = \epsilon L$ for dislocation of size $L$ and for relative strain $\epsilon$, the seismic moment $M_0$ can be written with the following form:

$$M_0 = \mu \epsilon L^3, \tag{5}$$

and the seismic moment of the calibration event, for which we introduce the dislocation size $L_{min}$, will be expressed as

$$m_0 = \mu \epsilon L_{min}^3. \tag{6}$$

We will substitute the expressions (4), (5), and (6) into (3) and obtain

$$M = \frac{2}{3}(\lg M_0 - \lg m_0) = \frac{2}{3}\lg \frac{M_0}{m_0} = 2\lg \frac{L}{L_{min}}. \tag{7}$$

Note that the obtained definition of magnitude (in Bells) in terms of the dislocation size does not include the parameters of the medium and does not depend on the units of their measurement of dislocation size. We have obtained a geometric definition of the magnitude. Then, taking into account the expressions (2) and (7), the probability distribution function of dislocation sizes takes the power-law form depending only on geometric characteristics,

$$P(L) = 1 - \frac{N}{N_{total}} = 1 - 10^{-bM} = 1 - \left(\frac{L}{L_{min}}\right)^{-2b}. \tag{8}$$

Differentiating expression (8) by the dislocation size $L$, we find the probability density

$$p(L) = \frac{dP(L)}{dL} = \frac{2b}{L_{min}}\left(\frac{L}{L_{min}}\right)^{-2b-1} = \frac{2b}{L_{min}} \cdot r^{-2b-1}, \tag{9}$$

then the increment of the function (8) can be represented as follows:

$$\Delta P(L) = p(L)\Delta L. \tag{10}$$

The largest size of dislocations, denoted as $L_{max}$, is limited by the size of the seismic polygon and amounts to values of the order of $10^2$ km. The corresponding maximum value of the magnitude $M_{max}$ is assumed to be equal to 9 [6]. Then, using the ratio (7), we get the value $L_{min} \sim 1$ meter.

If the dislocation size is $L_{min}$, then the magnitude takes the value $M = 0$ (7). Then, the considered range of dislocation sizes $L_{min} < L < L_{max}$ corresponds to an extended range of magnitudes $0 < M < 9$ compared to the definition (2), where $1 < M < 9$. Magnitudes from the range $0 < M < 1$ are registered in geoacoustics [9–12] and make a high-frequency contribution to the energy of a seismic event; in this regard, we do not exclude them from consideration.

The Gutenberg–Richter law in power-law form will be considered in the range of magnitudes $0 < M < 9$ as the average approximation (mean) of the energy spectrum of seismoacoustic oscillations. Curvatures from the power-law in narrower spectral ranges are considered as non-stationary [13,14]. The instability conditions of the power-law distribution will be described in detail hereinafter.

The magnitude step in seismic catalogs is $\Delta M = 0.1$, i.e., one deciBell. In this case, the partition of the interval $L_{min} < L < L_{max}$ of dislocation size will not be equidistant (uniform). We will make a uniform partition of the interval $L_{min} < L < L_{max}$ with the

step $\Delta L = L_{min}$ to form a generalized harmonic series of event recurrence frequencies. The number $k$ of partition intervals is equal to

$$k = \frac{L_{max}}{L_{min}} = 10^{4.75} \approx 10^5.$$

In each $\Delta L_r = L_{min}$, $L_{min} = 1$ in $r$ variable, we have the number $\Delta N_r$ of events, which is obtained by taking into account the expressions (10) and (9),

$$\Delta N_r = N_{total}\Delta P(L_r) = N_{total}p(L_r)\Delta L_r = N_{total}p(rL_{min})\Delta L_r = N_{total} \cdot 2b \cdot r^{-2b-1},$$

whence it follows that the event recurrence frequency $\omega_r$ for the observation period $T$ has a power-law distribution,

$$\omega_r = \frac{\Delta N_r}{T} = 2b\Omega \cdot r^{-2b-1}, \ r = 1,\ldots,k, \ \Omega = \frac{N_{total}}{T}, \tag{11}$$

where $\Omega$ is the total frequency of events. Therefore, the sum of the sequence $\omega_r$ (11) will be

$$\sum_{r=1}^{k} \omega_r = 2b\Omega \sum_{r=1}^{k} r^{-2b-1} = \Omega. \tag{12}$$

Note that, for large $k$ in our model we can take $k \to \infty$.

We supplement Equation (1) with a power-law distribution (11). The dimensionless expressions $\lambda_r dt^\nu$ resulting from Equation (1), can be represented in the form $\lambda_r dt^\nu = (\omega_r)^\nu dt^\nu = (\omega_r dt)^\nu$ in accordance with dimensional considerations for fractional $\lambda_r$ and non-fractional $\omega_r$ frequencies. Then, the positive fractional rates $\lambda_r$ will be associated with the frequencies $\omega_r$ (11) of the recurrence of random events by the ratio

$$\lambda_r = (\omega_r)^\nu = (2b\Omega \cdot r^{-2b-1})^\nu, \tag{13}$$

and the $\Lambda$-parameter, i.e., the fractional decay rate, will be represented by the sum of the sequence (13):

$$\Lambda = \sum_{r=1}^{k} \lambda_r = (2b\Omega)^\nu \sum_{r=1}^{k} r^{-(2b+1)\nu}. \tag{14}$$

## 3. Critical Indices and Process Instability

In this section, we will consider the role of the CFPP temporal non-locality defined by the fractional parameter $\nu$.

Firstly, according to equations (1) with the distribution (13) and taking into account the sum (14), the parameter

$$\Lambda^{1/\nu} = 2b\Omega \left( \sum_{r=1}^{k} r^{-(2b+1)\nu} \right)^{1/\nu} \tag{15}$$

is the decay rate of the initial and all subsequent states, which depends on the hereditarity parameter $\nu$. The total frequency $\Omega$ (11) determines the dimension and value of the rate (15).

Secondly, the probability distributions of the first-passage times for each scale $r$ [4], taking into account the ratios (13), can be presented in the following form

$$P(t) = \lambda_r t^\nu E_{\nu,\nu+1}(-\lambda_r t^\nu) = (\omega_r t)^\nu E_{\nu,\nu+1}(-(\omega_r t)^\nu), \ t \geq 0, \ r = 1, 2, \ldots, k, \tag{16}$$

where $E_{\nu,\nu+1}(x)$ is the Mittag–Leffler function, i.e., a fractional exponential function. The distributions (16) define the non-local correlations of random events over time. Thus, the CFPP has the property of «memory» manifested in the statistical dependence of events and delayed relaxation, which occur as a result of medium hardening.

Thirdly, the statistical characteristics of the distribution (13), such as the zero, first, and second moments will depend on the $\nu$ parameter. And the types of divergences of these characteristics define three types of instability of the CFPP (1) with (13).

The zero moment of the distribution (13) is the $\Lambda$ (14), i.e., the sum of the fractional frequencies or fractional rates $\lambda_r$. The first type of the CFPP instability is associated with the partial sum $S_k$ divergence of the generalized harmonic series in distribution, (13)

$$S_k = \sum_{r=1}^{k} r^{-(2b+1)\nu}. \tag{17}$$

A significant range of $L$ determines large values $k$, and we can put $k \to \infty$. Therefore, the partial sums (17) satisfy the condition

$$S_\infty = \lim_{k \to \infty} S_k = \zeta\big((2b+1)\nu\big), \tag{18}$$

where $\zeta(x)$ is the Riemann zeta function. The property of the Riemann zeta function $\lim_{x \to 1} \zeta(x) = \infty$ (divergence of the series) gives

$$(2b+1)\nu = 1.$$

Therefore, the value of the critical index will be

$$\nu_0 = \frac{1}{2b+1}.$$

If $\nu > \nu_0$, then the expression (18) converges and the CFPP is in a subcritical regime, otherwise the series diverges and the CFPP goes into critical ($\nu = \nu_0$) or supercritical ($\nu < \nu_0$) regimes.

Provided that the zero moment is finite, we can discuss two other types of the CFPP instability, which are related to the first and second moments of the distribution (13), which are determined up to a factor of $(2b\Omega)^\nu$ by partial sums:

$$S_{k,p} = \sum_{r=1}^{k} r^{-(2b+1)\nu+p}, \; p = 1, 2. \tag{19}$$

The mean $\mathbf{E}(t)$ of the CFPP (first moment), which is the mean sum of the CFPP changes and defines the mean total deformation, is represented by the expression [4]

$$\mathbf{E}(t) = S_{k,1}(2b\Omega \cdot t)^\nu / \Gamma(\nu+1), \quad t \geq 0, \tag{20}$$

and the variance $\mathbf{Var}(t)$ of the CFPP (CFPP second centered moment), which is the measure of fluctuations and defines the mean energy $\mathbf{W}(t)$ of the CFPP as $\mathbf{W}(t) = \mathbf{Var}(t) + \mathbf{E}^2(t)$, is expressed in the form [4]

$$\mathbf{Var}(t) = S_{k,2}(2b\Omega \cdot t)^\nu / \Gamma(\nu+1) + \big(S_{k,1}(2b\Omega \cdot t)^\nu\big)^2 Z(\nu), \quad t \geq 0, \tag{21}$$

where

$$Z(\nu) := \frac{1}{\nu}\left( \frac{1}{\Gamma(2\nu)} - \frac{1}{\nu\Gamma^2(\nu)} \right),$$

and $\Gamma(x)$ is the gamma function. If $\nu = 1$, then we get a simple Poisson process with the variance $\mathbf{Var}(t)$ (21) proportional to the mathematical expectation (mean) $\mathbf{E}(t)$ (20). If $0 < \nu < 1$, then the variance $\mathbf{Var}(t)$ includes a non-zero second term proportional to the square of the mean, which can be considered a manifestation of the random events coherence and as a result of the non-locality over time («memory») of the CFPP.

Note that the partial sums (19) (equal to moments of the distribution (13) without factor of $(2b\Omega)^\nu$) determine the rates of growth of the CFPP moments $\mathbf{E}(t)$ (20) and $\mathbf{Var}(t)$ (21), which define the mean deformations, variance, and mean energy of the seismic process, the instabilities of which will arise depending on the divergence of the partial sums (19).

The existence conditions of $S_{k,p}$ (19) in $k \to \infty$ are derived similarly as for $S_k$ (17)

$$S_{\infty,p} = \lim_{k \to \infty} S_{k,p} = \zeta\big((2b+1)\nu - p\big), \qquad (22)$$

and, based on the Riemann zeta function property, we obtain critical ratio

$$(2b+1)\nu - p = 1,$$

from which we find critical indices

$$\nu_p = \frac{1+p}{(2b+1)}, \ p = 1,2. \qquad (23)$$

We will also use the expression (23) to calculate $\nu_0$, taking $p = 0$.

The condition for the hereditary parameter $\nu > \nu_p$ of the power-law distribution (13) of event frequencies allows us to find stable statistical moments that set finite rates describing the process of seismic deformations. In the case $\nu < \nu_p$, the CFPP goes into a non-stationary regime as a result of instability [13,14]. Further, we will apply theoretical conclusions to study the characteristics of the deformation process based on the seismic catalog data.

## 4. Calculation of the Distribution Parameters of the Events Recurrence Frequency Based on Seismic Data

It should be noted that, in practice, to calculate the characteristics of the deformation process based on the hereditary model, it is possible to use the values of classes or magnitudes accepted in catalogs [8], instead of a geometric description in terms of dislocation sizes $L$. To determine the state of the deformation process, the parameters $b$-value, $\nu$, and $\nu_p$ are calculated in the interval of classes, where a power-law distribution (11) of event recurrence frequencies $\omega_r$ is performed. A comparison of the parameter $\nu$ averaged over the class interval used with critical indices $\nu_p$ allows us to conclude about the state of the deformation process.

### 4.1. Calculation of b-Value

In Section 3, an analytical expression (11) for the frequencies $\omega_r$ of events recurrence was derived based on the Gutenberg–Richter law (2), from which it follows that the distribution of the frequencies $\omega_r$ is defined by the parameter $b$ of the Gutenberg–Richter law. To calculate the $b$-value, we will also use the law (2) in the logarithmic form

$$\lg N = a - bM. \qquad (24)$$

We determined $b$-value for the earthquake catalog of the Kamchatka Branch of the Geophysical Survey of the Russian Academy of Sciences for the period from 1 January 1962 to 31 December 2002 for the Kuril–Kamchatka island arc subduction zone (area $46°$–$62°$ N, $158°$–$174°$ E) [8]. The catalog size is equal to $n = 79{,}282$ of earthquakes. The energy of a seismic event in the catalog is defined by the energy class, which is determined to the nearest tenths. The catalog contains events from 4.1 to 16.1 of energy classes [8].

Based on the obtained distribution of the earthquakes number depending on their energy class (magnitude), we concluded that the sample of earthquakes is representative for values of energy classes exceeding 8.3 (Figure 1a) (see file StatisticsGRlaw_n=79282.csv in Supplementary Materials). The size of a representative sample of earthquakes is $n = 46{,}917$. The interval of the energy classes included in this sample is equal to $[8.3, 16.1]$. The logarithmic form of the empirical Gutenberg–Richter law is represented in Figure 1b by dot graph. Based on the empirical dependence (Figure 1b), a rough estimate can be made that the linear part is located on the interval of classes $K \in [9, 13]$ (of magnitudes $M \in [2.5, 5.5]$). There is a nonlinearity in the interval $K < 9$ and a break in the linear part in the vicinity of the value $K = 13$.

We will find the best approximation of the linear part of the logarithmic Gutenberg–Richter law using the LSM. The estimation of the intervals boundaries of the $a$ and $b$

parameter changes is made based on the empirical dependence for the logarithmic form of the Gutenberg–Richter law (Figure 1b).

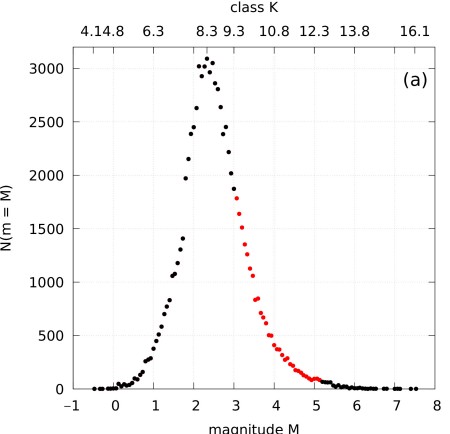 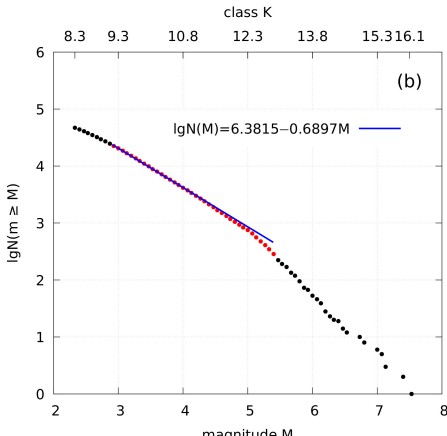

**Figure 1.** (**a**) Distribution of the earthquakes number depending on their energy class (magnitude) and (**b**) its logarithmic form for classes $K \in [8.3, 16.1]$. The dot graphs (black dots) are the distribution and its logarithmic form, where the approximation interval $K \in [9.2, 12.9]$ is highlighted with red dots; the blue graph is an approximation of the linear part of logarithmic form of the distribution.

The empirical Gutenberg–Richter law is approximated by the exponential function (2) (nonlinear regression). The values of the *a* and *b* parameters were determined by iterating over the values in increments equal to $h = 0.001$ from the intervals selected for them. For each pair of parameters values the approximation error $\varepsilon$, for which constraints $1\% < \varepsilon < 10\%$ was accepted, and the correlation index *R* were calculated. Then, the classes were sequentially excluded from the beginning and end of the interval $K \in [8.3, 16.1]$ ($M \in [2.33, 7.53]$) until the approximation error $\varepsilon$ reaches a minimum in the constraints accepted for it, provided that the value of the correlation index *R* is the maximum.

Also, at each step of the above algorithm, the empirical logarithmic Gutenberg–Richter law (Figure 1b) was approximated by a linear function (24) (linear regression) and its statistical characteristics were calculated. For the approximation error $\varepsilon$ of the logarithmic law (24), the following limitations are accepted $1\% \leq \varepsilon \leq 2\%$.

If an interval of classes is found upon which the accepted conditions for exponential approximation are executed, and the approximation error of the Gutenberg–Richter logarithmic law is minimal and satisfies the accepted constraints, then this interval of classes is chosen as the approximation interval.

The calculation results are presented in Table 1. At the approximation interval, the value of the correlation index $R = 0.9857$ of the nonlinear regression is close to one, which indicates a practically functional relationship of empirical data. The statistical significance of the regression equations as a whole is estimated using the Fischer criterion (*F*-test) $F(\alpha, k_1, k_2)$ at the significance level $\alpha = 0.05$ with degrees of freedom $k_1 = m - 1 = 1$ and $k_2 = k - m = k - 2$, where *k* is the number of classes in the considered interval of approximation and *m* is the number of regression parameters ($m = 2$). The empirical values of the statistics *F* are given in the Table 1 and exceed the critical value $\tilde{F} = F(0.05, 1, 42) = 4.08$. Therefore, at a given significance level $\alpha = 0.05$, we recognize the statistical significance of the regression equations as a whole, both nonlinear (Table 1, first row) and linear (Table 1, second row). Thus, for further calculations, we can use the found values of the *a* and *b* parameters of the Gutenberg–Richter law.

**Table 1.** Parameters of the Gutenberg–Richter law and statistical characteristics of approximating functions $F(X)$.

| $F(X)$ | $[K_1, K_2]$ [1] | $k$ | $N_{total}$ | $a$ | $b$ | $R$ | $F$ | $\tilde{F}$ | $\varepsilon, \%$ |
|---|---|---|---|---|---|---|---|---|---|
| $10^{a-bX}$ | [9.2, 12.9] | 38 | 22,230 | 6.3815 | 0.6897 | 0.9857 | 1233 | 4.08 | 1.658 |
| $a - bX$ | | | | | | 0.8567 | 99 | | 1.687 |

[1] Approximation interval (for magnitudes respectively equal to $[M_1, M_2] = [2.93, 5.40]$).

### 4.2. Distributions of the First-Passage Times

The distributions of the waiting times for the first movement or distribution of the first-passage times are compiled for each energy class that belongs to the approximation interval $K \in [9.2, 12.9]$ ($M \in [2.93, 5.40]$), i.e., we consider the distribution of times between pairs of neighboring events of the specified energy class.

The following calculation procedure was used. We selected the events of a fixed class $K_r, r = 1, \ldots, k$ from the catalog (the value $k$ is taken from the Table 1). For each pair of neighboring events of the class $K_r$, we calculated the length $l_t$ [days] of the time intervals between them. We have denoted by $T_{max}$ the greatest length of the interval. Then, the lengths $l_t$ of all intervals of time belong to the interval $(0, T_{max}]$. We divided the interval $(0, T_{max}]$ into intervals of one day length starting from zero. The number of intervals obtained is equal to $n_r = [T_{max}] + 1$, where $[T_{max}]$ is the integer part of the value $T_{max}$. We count the number of time intervals whose length $l_t$ satisfies following condition $t_i < l_t \leq t_{i+1}$, where $i = (0, 1, \ldots, n_r - 1)$, $t_0 = 0$ days, $t_{n_r} = T_{max}$ days, $t_{i+1} - t_i = 1$ day. Thus, we obtain an empirical distribution of the first-passage time for each energy class $K_r$ under consideration. For each such distribution, we form an empirical Cumulative Distribution Function (eCDF)— a step function (Figure 2a). In this case, function (16) is, respectively, considered a cumulative distribution function, which will be discussed below. The files with the calculation results for each class $K_r$ from the approximation interval are given in the Supplementary Materials. As an example, Figure 2a shows the eCDF of the first-passage times for class $K_{32} = 12.3$.

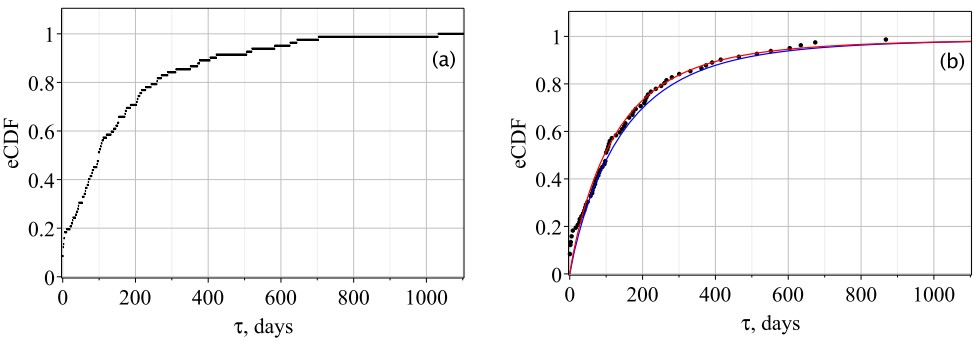

**Figure 2.** (**a**) The step function eCDF of the first-passage times for class $K_{32} = 12.3$. (**b**) The eCDF approximation by the function (16) for class $K_{32} = 12.3$, the dot graph (black dots in figure (**b**)) is eCDF (the corresponding relative frequency is mapped to the middle of each interval), the blue graph is one-parameter approximation, the red graph is two-parameter approximation.

### 4.3. Approximation of the First-Passage Times Distributions

We use probability function (16) to approximate the eCDF of the first-passage times. To approximate the eCDF (Figure 2a) using the LSM, we consider the point definition of the function (Figure 2b), where the corresponding relative frequency is mapped to the middle of each interval. The approximation was carried out for both empirical and calculated values of the frequency $\omega_r$ [$day^{-1}$] of events recurrence of the class $K_r$. In the first case, the frequency $\omega_r$ [$day^{-1}$] was calculated based on the Gutenberg–Richter law. The value of the parameter $\nu = \nu_r$ of the approximation function (16) was found on a

given interval $0 < \nu \leq 1$ (1) by iterating with increment of 0.001, based on the condition of minimizing the approximation error $\varepsilon$ (Table 2, columns 4–7). In the second case, we used a two-parameter approximation by the parameters $\omega_r \, [day^{-1}]$ and $\nu = \nu_r$ of the function (16). The choice of change interval of the frequency $\omega_r$ was determined by the largest and smallest empirical values of frequencies (Table 2, column 5). The $\omega_r$ and $\nu_r$ parameter values were found iteratively in increments of 0.001 under the condition of a minimum approximation error $\varepsilon$ (Table 2, columns 8–11).

**Table 2.** Parameters of first-passage times distributions.

| | | | Approximation by a Function (16) | | | | | | | |
| | | | One-Parameter | | | | Two-Parameter | | | |
| $K_r$ | $M_r$ | $n_r$ | RSS | $\omega_r, day^{-1}$ | $\nu_r$ | $\varepsilon, \%$ | RSS | $\omega_r, day^{-1}$ | $\nu_r$ | $\varepsilon, \%$ |
| 1 | 2 | 3 | 4 | 5 | 6 | 7 | 8 | 9 | 10 | 11 |
| 9.2 | 2.93 | 57 | 0.102 | 0.135 | 0.961 | 4.71 | 0.025 | 0.182 | 0.891 | 2.33 |
| 9.3 | 3.0 | 57 | 0.067 | 0.125 | 0.974 | 3.93 | 0.031 | 0.152 | 0.925 | 2.59 |
| 9.4 | 3.07 | 57 | 0.085 | 0.119 | 0.962 | 4.36 | 0.034 | 0.149 | 0.905 | 2.76 |
| 9.5 | 3.13 | 58 | 0.099 | 0.109 | 0.958 | 4.71 | 0.039 | 0.139 | 0.897 | 2.97 |
| 9.6 | 3.2 | 62 | 0.105 | 0.101 | 0.959 | 4.71 | 0.033 | 0.129 | 0.895 | 2.64 |
| 9.7 | 3.27 | 67 | 0.109 | 0.09 | 0.962 | 4.62 | 0.036 | 0.114 | 0.901 | 2.65 |
| 9.8 | 3.33 | 67 | 0.102 | 0.084 | 0.954 | 4.55 | 0.032 | 0.105 | 0.895 | 2.53 |
| 9.9 | 3.4 | 77 | 0.095 | 0.075 | 0.961 | 4.08 | 0.043 | 0.09 | 0.914 | 2.76 |
| 10.0 | 3.47 | 71 | 0.108 | 0.071 | 0.955 | 4.65 | 0.063 | 0.083 | 0.907 | 3.54 |
| 10.1 | 3.53 | 83 | 0.077 | 0.056 | 0.960 | 3.70 | 0.036 | 0.064 | 0.92 | 2.51 |
| 10.2 | 3.6 | 84 | 0.074 | 0.057 | 0.968 | 3.57 | 0.029 | 0.065 | 0.928 | 2.24 |
| 10.3 | 3.67 | 91 | 0.112 | 0.048 | 0.960 | 4.29 | 0.035 | 0.056 | 0.91 | 2.38 |
| 10.4 | 3.73 | 93 | 0.245 | 0.045 | 0.932 | 6.31 | 0.045 | 0.06 | 0.849 | 2.7 |
| 10.5 | 3.8 | 95 | 0.133 | 0.041 | 0.969 | 4.64 | 0.043 | 0.049 | 0.918 | 2.64 |
| 10.6 | 3.87 | 103 | 0.214 | 0.034 | 0.942 | 5.79 | 0.051 | 0.042 | 0.88 | 2.83 |
| 10.7 | 3.93 | 103 | 0.163 | 0.033 | 0.959 | 5.12 | 0.081 | 0.039 | 0.91 | 3.61 |
| 10.8 | 4.0 | 107 | 0.257 | 0.027 | 0.922 | 6.53 | 0.089 | 0.034 | 0.858 | 3.83 |
| 10.9 | 4.07 | 109 | 0.28 | 0.025 | 0.947 | 6.75 | 0.041 | 0.031 | 0.888 | 2.6 |
| 11.0 | 4.13 | 113 | 0.331 | 0.025 | 0.922 | 7.19 | 0.041 | 0.033 | 0.865 | 3.76 |
| 11.1 | 4.2 | 115 | 0.395 | 0.021 | 0.880 | 8.01 | 0.079 | 0.028 | 0.817 | 3.59 |
| 11.2 | 4.27 | 112 | 0.357 | 0.018 | 0.907 | 7.97 | 0.059 | 0.023 | 0.852 | 3.25 |
| 11.3 | 4.33 | 113 | 0.301 | 0.019 | 0.926 | 7.17 | 0.103 | 0.024 | 0.873 | 4.19 |
| 11.4 | 4.4 | 112 | 0.144 | 0.016 | 0.898 | 5.38 | 0.044 | 0.018 | 0.886 | 4.05 |
| 11.5 | 4.47 | 104 | 0.405 | 0.015 | 0.845 | 9.15 | 0.101 | 0.019 | 0.797 | 4.57 |
| 11.6 | 4.53 | 100 | 0.225 | 0.012 | 0.936 | 7.27 | 0.141 | 0.013 | 0.91 | 5.75 |
| 11.7 | 4.6 | 95 | 0.505 | 0.011 | 0.825 | 11.18 | 0.023 | 0.021 | 0.784 | 4.07 |
| 11.8 | 4.67 | 89 | 0.451 | 0.01 | 0.861 | 10.74 | 0.112 | 0.014 | 0.822 | 5.35 |
| 11.9 | 4.73 | 79 | 0.353 | 0.009 | 0.835 | 10.21 | 0.036 | 0.012 | 0.85 | 5.4 |
| 12.0 | 4.8 | 76 | 0.529 | 0.008 | 0.803 | 13.1 | 0.055 | 0.012 | 0.818 | 4.22 |
| 12.1 | 4.87 | 73 | 0.576 | 0.007 | 0.772 | 14.03 | 0.08 | 0.01 | 0.775 | 5.21 |
| 12.2 | 4.93 | 61 | 0.21 | 0.006 | 0.915 | 9.31 | 0.113 | 0.007 | 0.886 | 6.83 |
| 12.3 | 5.0 | 68 | 0.133 | 0.006 | 0.920 | 7.06 | 0.089 | 0.007 | 0.909 | 5.76 |
| 12.4 | 5.07 | 65 | 0.666 | 0.006 | 0.751 | 15.45 | 0.04 | 0.01 | 0.766 | 5.95 |
| 12.5 | 5.13 | 55 | 0.644 | 0.006 | 0.755 | 16.42 | 0.033 | 0.011 | 0.749 | 4.92 |
| 12.6 | 5.2 | 49 | 0.273 | 0.005 | 0.787 | 12.2 | 0.043 | 0.007 | 0.791 | 6.04 |
| 12.7 | 5.27 | 51 | 0.217 | 0.004 | 0.896 | 10.55 | 0.075 | 0.005 | 0.882 | 6.21 |
| 12.8 | 5.33 | 47 | 0.108 | 0.004 | 0.880 | 7.92 | 0.051 | 0.005 | 0.858 | 5.46 |
| 12.9 | 5.4 | 50 | 0.428 | 0.004 | 0.868 | 14.85 | 0.032 | 0.006 | 0.883 | 5.32 |

It follows from the results of the approximation that the two-parameter approximation is more accurate. The one-parameter approximation for some classes $K_r \geq 11.7$ gives errors $\varepsilon$ exceeding 10%. Therefore, for further calculations of the model parameters, we will use the values of frequency $\omega_r$ and exponent $\nu_r$ of the fractional derivative obtained by the two-parameter approximation method.

The values $\nu_r$ of the exponent $\nu$ of the fractional derivative obtained from the results of both one- and two-parameter approximations of eCDF for event streams of the considered

energy classes vary from 0.74 to 0.98. It is necessary to note the tendency to decrease the values of the exponent $\nu$ with an increase in the values of energy class $K_r$. This indicates the temporal non-locality of the seismic process, «memory» effects, statistical dependence of events, and delayed relaxation that occur as a result of the hardening of the medium.

## 5. Results and Discussion

The results obtained in the previous Section 4 allow us to calculate at the approximation interval the values characterizing the CFPP (1) with (13) of the order of $k$ with integer random state changes by a value $r = 1, \ldots, k$ [3,4]:

1. The hereditarity parameter or the average of the exponent $\nu$ of the fractional derivative of the CFPP is calculated on the values in column 10 of the Table 2

$$\frac{1}{k} \sum_{r=1}^{k} \nu_r = \nu = 0.8675.$$

   According to the value of this parameter, we can conclude that the considered seismic process has «memory», so random events of deformation changes cannot be considered independent.

   Since $\nu < 1$, the distributions (16) define the delayed relaxation of strains, which are associated with the hardening of the deformable medium and the accumulation of elastic energy, which may be the reason for the activation of the process.

2. The fractional decay rate of CFPP states is determined by the parameter $\Lambda \left[ day^{-\nu} \right]$ (13), (14), which is represented as follows:

$$\Lambda = \sum_{r=1}^{k} \lambda_r = \sum_{r=1}^{k} (\omega_r)^{\nu}.$$

   The $\Lambda$-value equal to the zero moment is calculated on the values in column 9 of Table 2 and in item 1,

$$\Lambda = 2.6401 \ day^{-0.8675} = (3.0622/day)^{0.8675}.$$

   Then, the decay rate of the initial and all subsequent states of the CFPP (1) with (13) is equal to

$$\Lambda^{1/\nu} = 3.0622 \ [day^{-1}].$$

3. The stability parameter of the CFPP takes the value

$$(2b + 1)\nu \approx 2.0641,$$

   where the $b$-value is taken from Table 1. This parameter defines the multiplicative effect of the scaling and hereditarity on the critical indices.

4. The values of the critical indices (23) are equal to

$$\nu_0 = 0.4230, \ \nu_1 = 0.8406, \ \nu_2 = 1.2608.$$

   A comparison of the $\nu$-parameter (item 1) with the critical indices $\nu_p$ ($p = 0, 1, 2$) shows that the seismic process is in a subcritical regime for the zero and first moments and in a supercritical regime for the second moment of distribution (13), which indicates the instability of deformation changes that can go into a non-stationary regime of the seismic process. The reason for this activation is indicated in item 1.

   This result means that the fractional decay rate $\Lambda = (2b\Omega)^{\nu} S_{\infty}$ of the seismic process, described by the parameters of the CFPP (14) and (18), and the average deformations (20), proportional to $S_{k,1}$ at $k = \infty$, are finite, and the divergence in the dispersion growth (21) caused by $S_{k,2}$ at $k = \infty$ leads to the instability of the process and its transition to a non-stationary regime considered in papers [13,14].

The anomalous growth of fluctuations caused by the hereditarity of the seismic process is represented in $\mathbf{Var}(t)$ (21) by the second term, which is proportional to the square of the mean $\mathbf{E}(t)$ (20), which is different to the first term and proportional to the mean $\mathbf{E}(t)$. If the first term in $\mathbf{Var}(t)$ (21) describes an ordinary deformation, then the second term describes an anomalous one caused by the consolidation of scales. This is a collective or induced coherent effect, the analogue of which in quantum optics is superluminescence, and in phase transition physics is explosive boiling. In the absence of hereditarity, this effect disappears, because if we take $\nu = 1$, then the second term of $\mathbf{Var}(t)$ in (21) will be zero based on the property of the gamma function $\Gamma(z + 1) = z\Gamma(z)$.

## 6. Conclusions

The specificity of criticality is determined by scaling and hereditarity. The first property of the seismic process is responsible for the consolidation of dislocation scales, and the second one is for the correlation of events at time intervals. Since the critical indices (item 4) depend on the product of the parameters $b$ and $\nu$, we can talk about the manifestations of the multiplicative effect of scaling and hereditarity in critical phenomena.

The slowing down of relaxations and the accumulation of energy are the reason for the catastrophic nature of critical phenomena. The delayed relaxations and the anomalous growth of fluctuations can be considered as a precursor of a catastrophe, the scenario of which is determined by the non-stationary regime of the Poisson process [13,14].

The seismic process is diverse and can have many representations. The analysis of seismic data based on the hereditarian criticality model showed the instability of the quasi-stationary and quasi-homogeneous regime of the seismic process. Further, the non-stationary and spatially inhomogeneous regimes of the seismic process are investigated in more detail than in [13], which are represented by foreshocks and aftershocks. Of particular interest in terms of the study of anomalous diffusion are chains of seismic events [14–20] with their random walk, fading, and Levy flights. It will be necessary to understand how anomalous diffusion is affected by the criticality of the process. The variety of types of fractional derivatives can be used to generalize the case of the Caputo derivative considered here and to determine their influence on the criticality of the process. A more general case than in Equation (1) is also interesting, when the hereditary parameter $\nu$ depends on $r = 1, 2, \ldots, k$, i.e., the event streams differ in terms of the «memory» properties. The processing of seismic data showed exactly this dependence.

**Supplementary Materials:** The following supporting information can be downloaded at: https://www.mdpi.com/article/10.3390/fractalfract7120890/s1, File StatisticsGRlaw_n=79282.csv: statistics of the Gutenberg-Richter law; Folder subsection 5.2: files with an empirical distribution of the first-passage time for each energy class $K_r = 9.2 \ldots 12.9$ under consideration.

**Author Contributions:** Conceptualization, B.S.; Formal analysis, O.S.; Methodology, B.S. and O.S.; Project administration, B.S. and O.S.; Software, O.S.; Validation, B.S. and O.S.; Visualization, O.S.; Writing—original draft, O.S.; Writing—review and editing, B.S. and O.S. All authors have read and agreed to the published version of the manuscript.

**Funding:** This research was funded by State task AAAA-A21-121011290003-0.

**Data Availability Statement:** The Geophysical Service of the Russian Academy of Sciences. Available online: http://www.gsras.ru/new/eng/catalog/ (accessed on 27 February 2022).

**Conflicts of Interest:** The authors declare no conflicts of interest.

## Abbreviations

The following abbreviations are used in this manuscript:

| | |
|---|---|
| CFPP | Compound Fractional Poisson Process |
| LSM | Least Squares Method |
| eCDF | Empirical Cumulative Distribution Function |

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
