# Peer review of "Fractional Criticality Theory and Its Application in Seismology"

_fractalfract, doi:10.3390/fractalfract7120890_

Round 1

Reviewer 1 Report

Comments and Suggestions for Authors

The paper is devoted to application of the power-law compound and time-fractional Poisson process for modelling critical phenomena, 

including seismic ones. Apart from theoretical considerations, the authors show the usefulness of this model in practice. 

The article can be useful from a practical point of view. Moreover, it fits within the scope of Fractal Fract. However, it requires a minor revision.

Remarks 

  1. Please briefly describe the other sections of the paper in Introduction.
  2. Subsection 2.1: Please describe a compound fractional Poisson process in more detail. In particular, please introduce the filtered probability space on which this process is defined and present the role of the parameter j. 
  3. Please outline further research plans in Conclusions.

Reviewer 2 Report

Comments and Suggestions for Authors

see attached file

Comments on the Quality of English Language

Some typos and little errors but looks overall ok.

Author Response

We thank the reviewer for his comment.
